# Neuron-specific knockouts indicate the importance of network communication to *Drosophila* rhythmicity

Matthias Schlichting, Madelen M Díaz, Jason Xin, Michael Rosbash*

Department of Biology, Howard Hughes Medical Institute, Brandeis University, Waltham, United States

**Abstract** Animal circadian rhythms persist in constant darkness and are driven by intracellular transcription-translation feedback loops. Although these cellular oscillators communicate, isolated mammalian cellular clocks continue to tick away in darkness without intercellular communication. To investigate these issues in *Drosophila*, we assayed behavior as well as molecular rhythms within individual brain clock neurons while blocking communication within the ca. 150 neuron clock network. We also generated CRISPR-mediated neuron-specific circadian clock knockouts. The results point to two key clock neuron groups: loss of the clock within both regions but neither one alone has a strong behavioral phenotype in darkness; communication between these regions also contributes to circadian period determination. Under these dark conditions, the clock within one region persists without network communication. The clock within the famous PDF-expressing s-LNv neurons however was strongly dependent on network communication, likely because clock gene expression within these vulnerable sLNvs depends on neuronal firing or light.
DOI: https://doi.org/10.7554/eLife.48301.001

*For correspondence:
rosbash@brandeis.edu

Competing interests: The authors declare that no competing interests exist.

## Introduction

Neuronal networks make myriad contributions to behavior and physiology. By definition, individual neurons within a network interact, and different networks also interact to coordinate specialized functions. For example, the visual cortex and motor output centers must coordinate to react properly to environmental changes. In a less immediate fashion, sleep centers and circadian clocks are intertwined to properly orchestrate animal physiology. The brain clock is of special interest: it not only times and coordinates physiology within neuronal tissues but also sends signals to the body to keep the entire organism in sync with the cycling external environment (*Mohawk et al., 2012*).

The small, circumscribed *Drosophila* clock network is ideal to address circadian communication issues. The comparable region in mammals, the suprachiasmatic nucleus, is composed of thousands of cells depending on the species. There are in contrast only 75 clock neurons per hemisphere in *Drosophila*. These different clock neurons can be divided into several subgroups according to their location within the fly brain. There are 4 lateral and three dorsal neuron clusters, which have different functions in controlling fly physiology (*Helfrich-Förster et al., 2007*).

The four small ventro-lateral neurons (sLNvs) are arguably the most important of the 75 clock neurons. This is because ablating or silencing these neurons abolishes rhythms in constant darkness (DD). They reside in the accessory medulla region of the fly brain, an important pacemaker center in many insects (*Helfrich-Förster, 1997*), and express the neuropeptide PDF. In addition, they are essential for predicting dawn (*Depetris-Chauvin et al., 2011*; *Grima et al., 2004*; *Nitabach et al., 2002*; *Stoleru et al., 2004*). A very recent study suggests that the sLNvs are also able to modulate the timing of the evening (E) peak of behavior via PDF (*Renn et al., 1999*; *Schlichting et al., 2019a*). The other ventral-lateral group, the four large-ventro-lateral neurons (lLNvs), also express PDF and

send projections to the medulla, the visual center of the fly brain; they are important arousal neurons (*Shang et al., 2008*; *Sheeba et al., 2008*). Consistent with the ablation experiments mentioned above, the absence of *pdf* function or reducing PDF levels via RNAi causes substantial arrhythmic behavior in DD (*Renn et al., 1999*; *Shafer and Taghert, 2009*).

Other important clock neurons include the dorso-lateral neurons (LNds), which are essential for the timing of the E peak and adjustment to long photoperiods (*Grima et al., 2004*; *Kistenpfennig et al., 2018*; *Stoleru et al., 2004*). Two other clock neuron groups, the lateral-posterior neurons (LPN) and a subset of the dorsal neurons (DN1s), were recently shown to connect the clock network to sleep centers in the fly central complex (*Guo et al., 2018*; *Guo et al., 2016*; *Lamaze et al., 2018*; *Ni et al., 2019*). The DN2 neurons are essential for temperature preference rhythms (*Hamada et al., 2008*), whereas no function has so far been assigned to the DN3s.

Despite these distinct functions, individual clock neuron groups are well-connected to each other. At the anatomical level, all lateral neuron clusters and even DN1 dorsal neurons send some of their projections into the accessory medulla, where they can interact. A second area of common interaction is the dorsal brain; only the lLNvs do not project there (*Helfrich-Förster et al., 2007*).

Several studies have investigated interactions between different clock neurons. Artificially expressing kinases within specific clock neurons causes their clocks to run fast or slow and also changes the overall free-running period of the fly, indicating that network signaling adjusts behavior (*Chatterjee et al., 2018*; *Collins et al., 2014*; *Dissel et al., 2014*; *Rieger et al., 2006*; *Yao et al., 2016*; *Yao and Shafer, 2014*). Similarly, speeding up or slowing down individual neurons is able to differentially affect behavioral timing in standard light-dark (LD) cycles (*Stoleru et al., 2005*; *Yao et al., 2016*). A high level of neuronal plasticity within the network also exists: axons of individual cells undergo daily oscillations in their morphology (*Fernández et al., 2008*), and neurons change their targets depending on the environmental condition (*Gorostiza et al., 2014*; *Chatterjee et al., 2018*).

How neuronal communication influences the fly core feedback loop is not well understood. The latter consists of several interlocked transcriptional-translational feedback loops, which probably underlie rhythms in behavior and physiology (*Hardin, 2011*). A simplified version of the core feedback loop consists of the transcriptional activators Clock (CLK) and Cycle (CYC) and the transcriptional repressors Period (PER) and Timeless (TIM). CLK and CYC bind to E-boxes within the *period* (*per*) and *timeless* (*tim*) genes (among other clock-controlled genes) and activate their transcription. After PER and TIM synthesis in the cytoplasm, they form a heterodimer and enter the nucleus toward the end of the night. There they interact with CLK and CYC, release them from their E-box targets and thereby inhibit their own transcription. All 75 pairs of clock neurons contain this canonical circadian machinery, which undergoes daily oscillations in level. Indeed, the immunohistochemical cycling of PER and TIM within these neurons is a classic assay to visualize these molecular oscillations (*Menegazzi et al., 2013*).

Silencing PDF neurons stops their PER cycling, indicating an important role of neuronal firing in maintaining circadian oscillations. However, only two time points were measured, and the results were possibly confounded by developmental effects (*Depetris-Chauvin et al., 2011*; *Nitabach et al., 2002*). PDF neuron silencing also phase advances PER cycling in downstream neurons, suggesting that PDF normally serves to delay cycling in target neurons (*Wu et al., 2008*). This is consistent with experiments showing that PDF signaling stabilizes PER (*Li et al., 2014*). In addition, neuronal activation is able to mimic a light pulse and phase shift the clock due to firing-mediated TIM degradation (*Guo et al., 2014*).

To investigate more general features of clock neuron interactions on the circadian machinery, we silenced the majority of the fly brain clock neurons and investigated behavior and clock protein cycling within the circadian network in a standard light-dark cycle (LD) as well as in constant darkness (DD). Silencing abolished rhythmic behavior but had no effect on clock protein cycling in LD, indicating that the silencing affects circadian output but not oscillator function in a cycling light environment. Silencing similarly abolished rhythmic behavior in DD but with very different effects on clock protein cycling. Although protein cycling in the LNds was not affected by neuronal silencing in DD, the sLNvs dampened almost immediately. Interestingly, this differential effect is under transcriptional control, suggesting that some *Drosophila* clock neurons experience activity-regulated clock gene transcription. Cell-specific CRISPR/Cas9 knockouts of the core clock protein PER further suggests that network properties are critical to maintain wild-type activity-rest rhythms. Our data taken

together show that clock neuron communication and firing-mediated clock gene transcription are essential for high amplitude and synchronized molecular rhythms as well as rhythmic physiology.

## Results

To investigate the effects of clock network communication on fly behavior, we silenced most adult brain clock neurons using UAS-*Kir* (*Johns et al., 1999*). To this end, we used the *clk856*-GAL4 driver, which is expressed in most clock neurons (*Gummadova et al., 2009*) and first addressed locomotor activity behavior in a 12:12 LD cycle.

Both control strains show the expected morning and evening (M and E) anticipation increases, which are normal behavioral manifestations of clock function (*Figure 1A, C and G*). There is however no discernable activity anticipation in the silenced flies (*Figure 1G*). Only brief activity increases are visible, precisely at the day/night and night/day transitions (*Figure 1B*); these are startle responses (*Rieger et al., 2003*). Flies lacking PER show similar behavior (*per*[01] *Figure 1D* and *Figure 1H*).

To address possible developmental defects, we added *tub-GAL80ts* as an additional transgene to silence the clock network in an adult-specific manner. In this system, GAL80 is active at low temperatures (18°C) and inhibits GAL4 expression. By increasing the temperature to 30°C, GAL80 is inactivated, GAL4 is then functional and the *clk856* network silenced (*McGuire et al., 2003*).

At the low temperature, the controls and experimental lines show a typical wild-type bimodal activity pattern, which disappeared in experimental flies after switching to the high temperature (*Figure 1—figure supplement 1*). This effect was already visible on day one of high temperature exposure which shows that the *clk856>Kir* phenotype is not caused by defects during development. However, we cannot rule out chronic effects due to the prolonged period of network silencing.

We next compared the behavior to flies with silenced PDF neurons. Adult-specific silencing of the PDF neurons using the gene-switch system significantly advanced the timing of the E peak (*Figure 1E and F*), which reproduces previously published results (*Depetris-Chauvin et al., 2011*; *Nitabach et al., 2002*). However, the comparison with the *clk856* results shown above indicates that silencing the whole clock neuron network causes a much more severe behavioral phenotype than only silencing the PDF cells, that is network silencing completely abolishes rhythmic LD behavior, similar to clock mutant flies.

How does network silencing affect the circadian molecular feedback loop? To address this issue, we assayed PER as well as PDP1 protein levels in individual clock neuron clusters at four different times during the LD cycle. Both proteins show robust cycling: PER peaks at the end of the night (ZT0), and PDP1 peaks slightly earlier than PER as expected (*Hardin, 2011*) (*Figure 1I–1N*). This indicates that network silencing has no detectable effect on clock protein timing or cycling amplitude in LD. These data further suggest that either the different neuron clocks are self-sustained, comparable to the mammalian liver, or that light can drive rhythmic gene expression even in absence of neuronal communication.

To distinguish between these possibilities, we assayed behavior and molecular cycling in constant darkness (DD). Only 17% of the silenced flies were rhythmic, indicating that network silencing causes high levels of DD arrhythmicity (*Figure 2A*). To rule out developmental effects, we applied the *tub-GAL80ts* system as described above: 80 percent of the experimental flies were rhythmic at 18°C, but they were profoundly arrhythmic at 30°C with only two rhythmic flies (*Figure 2—figure supplement 1*). In contrast, adult-specific silencing of only the PDF neurons more weakly reduced rhythmicity (*Figure 2A*) and also caused a short period (*Figure 2B*), phenotypes that are essentially indistinguishable from those of the classical *pdf*[01] mutant (*Renn et al., 1999*).

To address why network silencing has such a profound effect, we assayed PER and PDP1 protein cycling after five days in constant darkness (DD5). As expected, all assayed clock neurons from control strains maintain robust and coordinated cycling in DD (*Figure 2C–H*); the sLNvs, LNds and DN1s peak slightly sooner than in LD, consistent with the slightly less than 24 hr circadian period in DD (*Figure 2B*).

In striking contrast, silencing the clock network causes clock protein cycling within the individual neuronal subgroups to differ strongly from each other, in amplitude and in phase. Clock protein cycling in the LNds is least affected by neuronal silencing and with little to no change in phase or amplitude, suggesting a robust and possibly self-autonomous clock in these neurons; see Discussion (*Figure 2D and G*). The sLNvs in contrast dampen and rapidly become arrhythmic, suggesting that

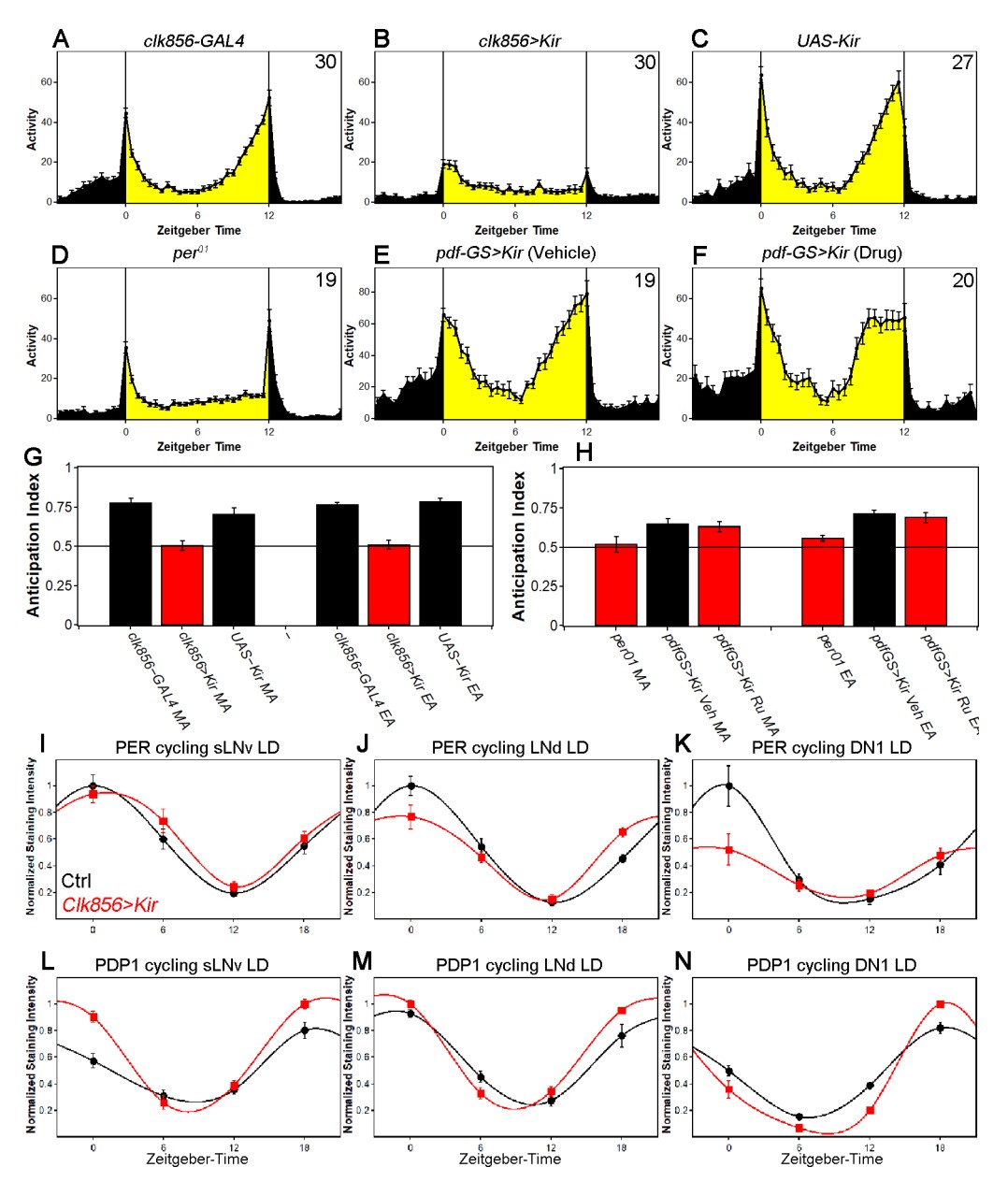

**Figure 1.** Silencing the clock network has differential effects on behavior and clock protein cycling in LD. (**A–C**) Silencing most of the clock network abolishes rhythmic LD behavior. GAL4 control (**A**) and UAS control (**C**) show bimodal activity patterns with M anticipation and E anticipation. Silenced flies (**B**) show no sign of anticipation neither in the morning nor in the evening. Flies show short activity increases at the transitions of day/night and night/day which are considered masking. Values in upper right-hand corner indicate the number of analyzed flies. (**D**) Behavior of *per*[01] flies in LD 12:12. *per*[01] mutants show behavior similar to *clk856>Kir* with no M anticipation, reduced E anticipation and short reactions to the light transitions. (**E–F**) Silencing PDF neurons alters LD behavior. (**E**) *PDF-GS>Kir* on Vehicle food does not express *Kir*. Flies show the typical bimodal activity with M and E anticipation peaks. The M peak is close to lights-on (ZT0) whereas the E peak is close to lights-off (ZT12). (**F**) Silencing the PDF neurons by adding RU 486 to the food causes an advanced E peak, similar to *pdf*[01] flies. Values indicate the number of analyzed flies. (**G**) Morning Anticipation (MA) and Evening Anticipation (EA) calculated from A-C. Both controls show values significantly above 0.5 (p<0.0001 for all) indicating prominent anticipation to both peaks. *Clk856>Kir* flies on the other hand show no signs of anticipation as indicated by anticipation indices indistinguishable from 0.5 (p>0.6937) (**H**) Morning Anticipation (MA) and Evening Anticipation (EA) calculated from D-E. *per*[01] flies show no MA (p=0.5744), whereas there is a slight increase of activity toward lights-off (p=0.0027).

*Figure 1 continued on next page*

*Figure 1 continued*

Silencing the PDF neurons did not abolish MA (p=0.0004) and showed a significantly bigger EA compared to *per$^{01}$* flies (p=0.0391). (I-K) PER protein cycling is largely unaffected by neuronal silencing in LD. PER cycling in control brains (black data points ± SEM, pooled GAL4 and UAS, n = 6–8) is highly synchronized with peak levels around ZT0. Silencing the clock network (red data points ± SEM, n = 5–7) had little effect on LD PER rhythms in sLNvs ($F_{(1,49)}$=0.93, p=0.3391) (I) and LNds ($F_{(1,49)}$=0.35, p=0.5547) (J). DN1s appear dampened after silencing but 2-way ANOVA shows no difference between control and experimental line ($F_{(1,36)}$=2.98, p=0.0930) (K). (L-N) PDP1 protein cycling is largely unaffected by neuronal silencing in LD. PDP1 cycling in control brains (black data points ± SEM, pooled GAL4 and UAS, n = 5) is highly synchronized with peak levels around ZT18. Silencing the clock network (red data points ± SEM, n = 5) slightly increased LD PDP1 rhythms in sLNvs ($F_{(1,32)}$=15.74, p=0.0004) (L) but no effect on LNds ($F_{(1,32)}$=2.71, p=0.1093) (M) and only a slight effect on DN1s ($F_{(1,32)}$=6.06, p=0.0194) (N).
DOI: https://doi.org/10.7554/eLife.48301.002

The following figure supplement is available for figure 1:

**Figure supplement 1.** Adult-specific silencing of most clock neurons causes arrhythmic LD behavior.
DOI: https://doi.org/10.7554/eLife.48301.003

---

these cells are rather weak oscillators and require network activity or light for proper molecular rhythms (*Figure 2C and F*). The DN1s also dampen but less strongly. They manifest low amplitude cycling, which is phase-advanced; this intermediate situation suggests a fast and somewhat network dependent clock in DN1s (*Figure 2E and H*). The DN2s were similar to the DN1s (data not shown). A comparable set of effects were observed in adult-specific silencing experiments (*Figure 2—figure supplement 2*).

To compare the contribution of network communication with the release of PDF to PER protein cycling, we also silenced only the PDF neurons in the adult (*Figure 2—figure supplement 2*). This slightly decreased the cycling amplitude in the sLNvs (p<0.01) but to a much smaller extent than silencing the entire brain clock (p<0.01). The LNds appeared unaffected by PDF neuron silencing, whereas the shape but not the amplitude of the PER cycling curve was altered in the DN1s (p=0.1628). These data indicate that PDF contributes to network synchrony but cannot explain all clock protein cycling changes in the *clk856>Kir* experiments.

To further address the molecular basis of the silencing dependence, we applied a fluorescent in-situ hybridization (fish) protocol to whole-mount *Drosophila* brains. Because *per* mRNA was undetectable, likely due to low expression within the clock network (data not shown and *Abruzzi et al., 2017*), we assayed *tim* mRNA cycling in LNds and in sLNvs as a proxy for clock gene transcription/ mRNA levels (*Figure 3A*). In control flies under LD conditions, *tim*-RNA cycles robustly in both sLNvs and LNds with a peak toward the beginning of the night as expected. In addition, clock network silencing had no effect on *tim* mRNA cycling amplitude or phase in LD, which parallels the protein cycling results (*Figure 3B and C*). In constant darkness (DD5), the controls show robust cycling in both sLNvs and LNds as expected, but silencing causes a profound decrease in *tim* mRNA signal in the sLNvs; the LNds cycle normally (*Figure 3D and E*). These data indicate a direct correlation between neuronal activity and *tim* RNA levels at least in the sLNvs and suggest that the silencing-mediated changes in clock protein cycling are in part transcriptional in origin.

Network silencing therefore reveals different levels of autonomy and endogenous speeds among clock neuron clusters. This leads to a drifting apart of the different subgroups from their usual well-synchronized and robust clock protein expression pattern. Interestingly, it appears that these phase differences are too big to re-establish coordinated rhythms after one week of silencing; there is no indication of rhythmic behavior upon lowering the temperature in the *tubGAL80ts* experiment (*Figure 2—figure supplement 3*).

The results to this point indicate that neuronal activity/communication is essential for rhythmicity as well as synchronized, high amplitude clock protein cycling in DD conditions. However, these results do not provide a hierarchy among the different groups, nor do they address a need for the circadian clock within these neurons. To distinguish between these possibilities and to develop a general knock-out strategy within the adult fly brain, we established a cell-specific CRISPR/Cas9 strategy to eliminate the circadian clock in individual clock neuron groups (*Figure 4A*). We applied the guide protocol introduced by *Port and Bullock (2016)* and cloned three guides targeting the coding sequence of *per* under UAS control and generated UAS-*per-g* flies. For a first experiment,

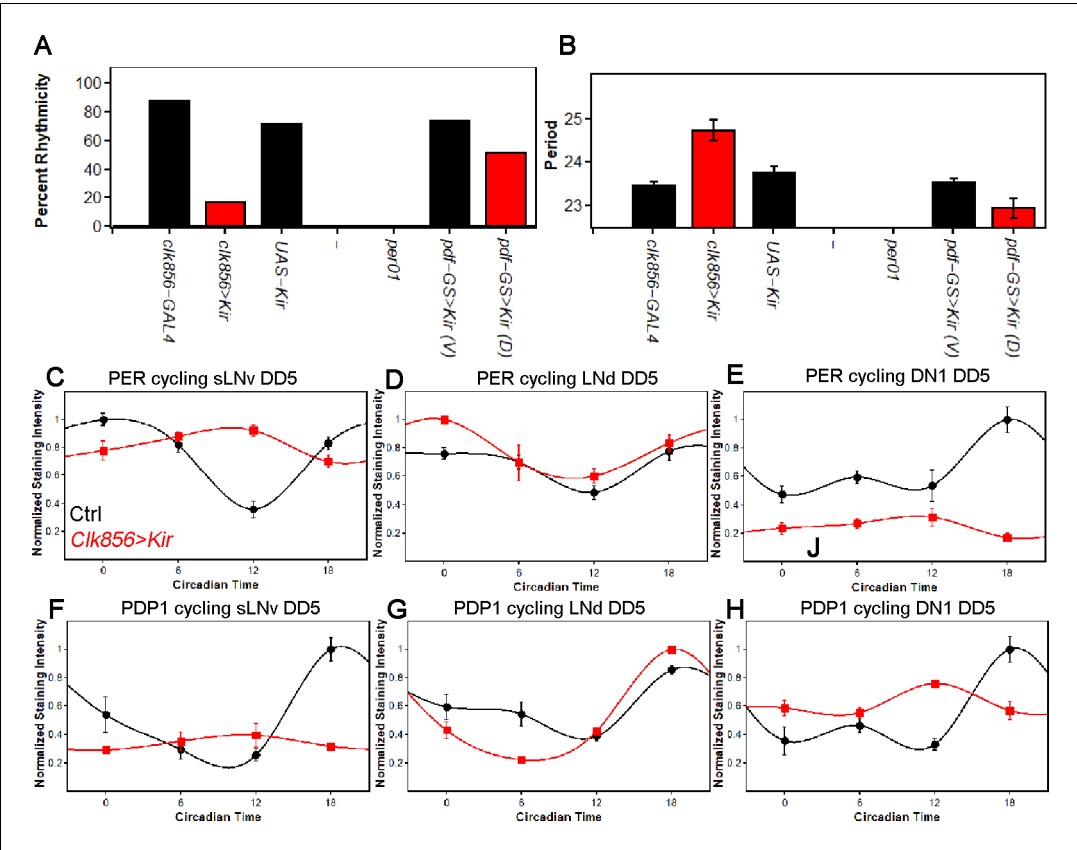

**Figure 2.** Silencing the clock network strongly affects behavior and molecular rhythms in DD. (**A**) Percentage of rhythmic flies in DD. Silencing most of the clock neurons significantly reduces rhythmicity to less than 20 percent, suggesting that clock neuron activity is essential for rhythmic behavior output. None of the $per^{01}$ flies were rhythmic as expected. Silencing the PDF neurons slightly decreased the level of rhythmicity. Number of flies analyzed same as shown in *Figure 1*. (**B**) Free-running period of rhythmic flies from (**A**). The few rhythmic flies of *clk856>Kir* show a significantly longer period ($F_{(2,48)}$=14.355 p<0.001, p<0.01 for UAS and GAL4 control). Adult-specific silencing of the PDF neurons caused a significant period shortening (p=0.0214). Number of flies analyzed same as shown in *Figure 1*. (**C–E**) PER protein cycling in DD5. PER cycling in control brains (black data points ± SEM, pooled GAL4 and UAS, n = 8–11) is highly synchronized with peak levels around CT18. Silencing the clock network (red data points ± SEM, n = 5) had variable effects on PER rhythms: The sLNvs (**C**) dampen strongly upon silencing (p<0.0001). The LNds (**D**) show no significant differences in cycling amplitude (p=0.8905) and the DN1s strongly dampen (p=0.0195), similar to sLNvs (**E**). (**F–H**) PDP1 protein cycling in DD5. PDP1 cycling in control brains (black data points ± SEM, pooled GAL4 and UAS, n = 5) is highly synchronized with peak levels around CT18. Silencing the clock network (red data points ± SEM, n = 5) had similar effects as observed in PER rhythms: The sLNvs (**F**) dampen strongly upon silencing (p=0.0006). The LNds (**G**) show a higher cycling amplitude than control brains (p<0.0001) and the DN1s strongly dampen (p=0.0013) and appear to have a phase-advanced PDP1 peak (**H**).

DOI: https://doi.org/10.7554/eLife.48301.004

The following figure supplements are available for figure 2:

**Figure supplement 1.** Adult-specific silencing of most clock neurons causes arrhythmic DD behavior.
DOI: https://doi.org/10.7554/eLife.48301.005
**Figure supplement 2.** Adult-specific silencing reproduces PER cycling profiles.
DOI: https://doi.org/10.7554/eLife.48301.006
**Figure supplement 3.** Removing neuronal silencing after 6 days in DD does not re-establish rhythmic behavior.
DOI: https://doi.org/10.7554/eLife.48301.007

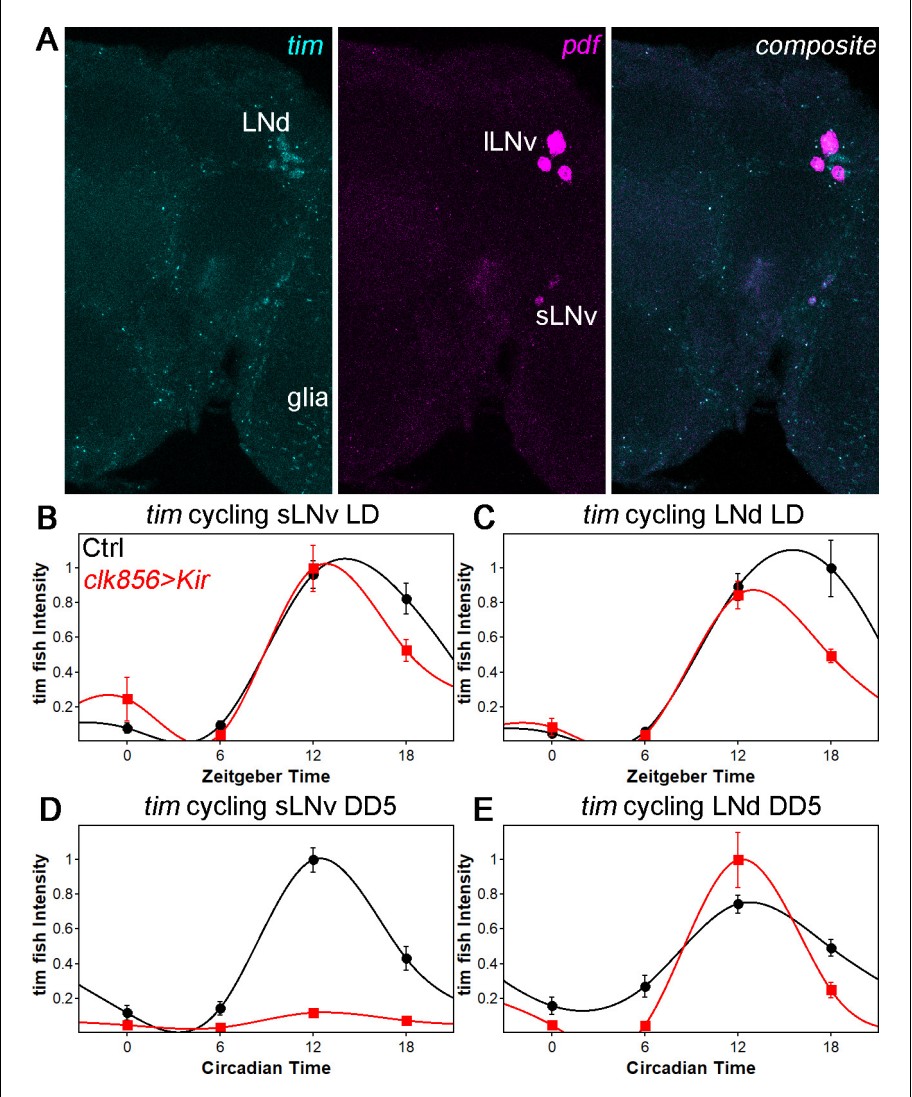

**Figure 3.** *tim* mRNA cycling in sLNvs and LNds shows similar trends as protein cycling as observed by FISH. (**A**) Representative brain showing *tim* (left panel), *pdf* (middle panel) and a composite of both channel (right panel). Tim probes label all clock neurons and glial cells, whereas the pdf probes only label sLNvs and lLNvs. (**B–C**) *tim* mRNA cycling in LD 12:12 in sLNvs (**B**) and LNds (**C**). Control flies (black data points ± SEM, pooled GAL4 and UAS, n = 10) show high amplitude cycling with peak levels at the beginning of the night. Silencing the clock network (red data points ± SEM, n = 5) has only little effect on cycling amplitude or timing in LD. (**D–E**) *tim* mRNA cycling in DD5 in sLNvs (**D**) and LNds (**E**). Control flies (black data points ± SEM, pooled GAL4 and UAS, n = 10) show high amplitude cycling with peak levels at the beginning of the night. Silencing the sLNvs (**D**) leads to an overall reduction of *tim* mRNA levels and a loss of rhythmicity. In the LNds (**E**) silencing did not decrease cycling amplitude or shifted peak mRNA expression.

DOI: https://doi.org/10.7554/eLife.48301.008

we expressed the *per*-guides and *Cas9* in most of the clock neuron network under *clk856* control and performed behavioral (*Figure 4B–4E*) and immunocytochemical (*Figure 4F–4H*) assays.

This PERKO strategy abolished M and E anticipation in LD behavior without affecting the startle responses (*Figure 4C* and *Figure 4E*), and it also reduced the level of DD rhythmicity to below 10%; this reproduced network silencing as well as the canonical *per$^{01}$* behavioral phenotypes (*Figures 1D* and *2A*). Not surprisingly perhaps given these robust phenotypes, immunohistochemistry indicates that the PERKO strategy works at more than 90% efficiency as determined by the loss of PER immunoreactivity in the lateral neuron clusters of the *clk856>Cas9 per-g* experiment (*Figure 4F–4H*). For

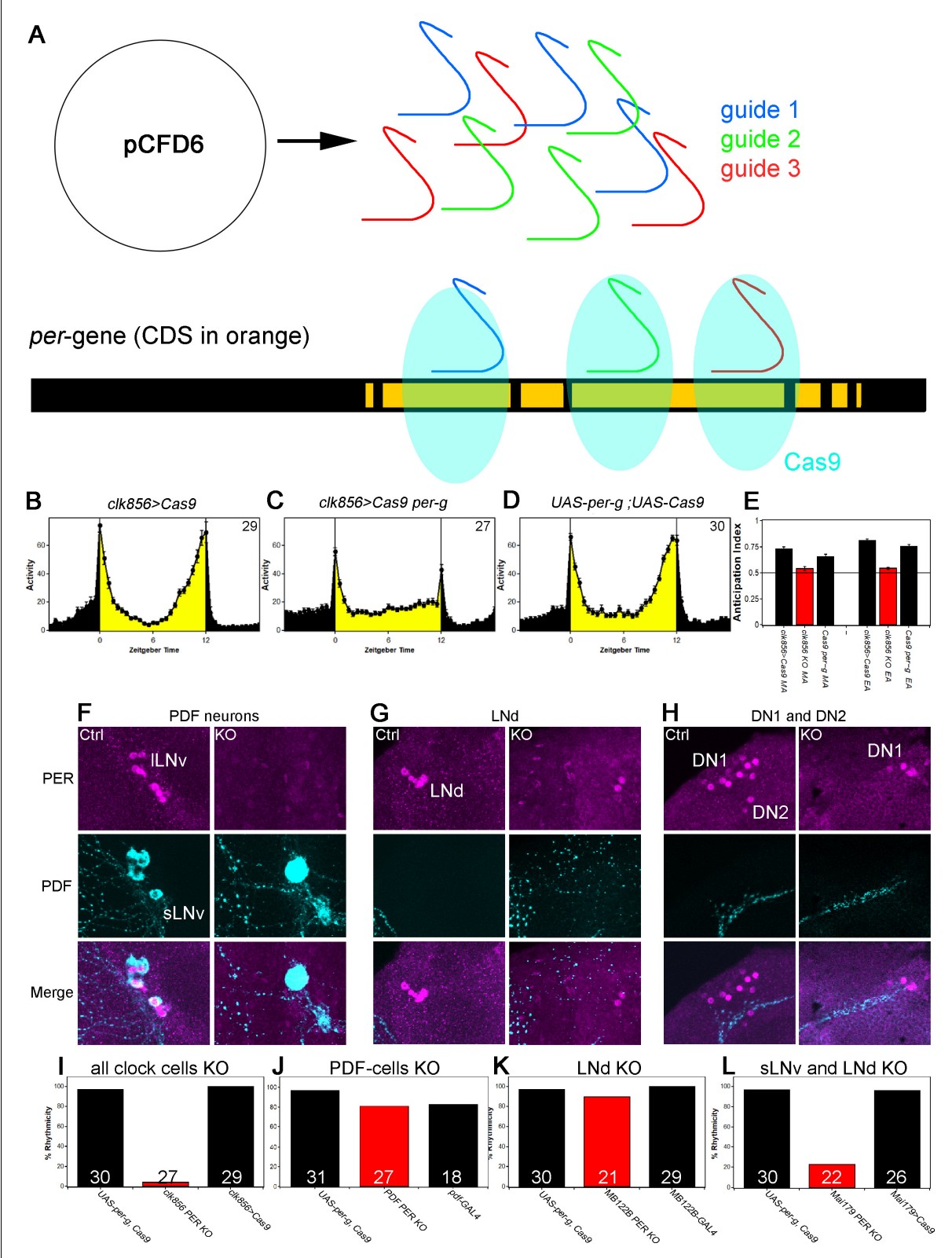

**Figure 4.** A clock in the LNds or the sLNvs can drive rhythmic behavior. (**A**) Schematic model of cell-specific knockout (KO) strategy. We generated a UAS-*per-g* line using the pCFD6 vector, allowing us to express three guides under the control of one UAS promoter (after *Port and Bullock, 2016*). We cloned three guides targeting the *per* CDS with guide one targeting the second exon shared by all transcripts and guides 2 and 3 targeting the 4th commonly shared exon. The guides will recruit the Cas9 protein and induce double-strand breaks and thereby cause mutations which lead to a non-
*Figure 4 continued on next page*

*Figure 4 continued*

functional protein. (B–E) Behavior of perKO using *clk856*-GAL4 in LD 12:12 reproduces *per01* phenotype. Flies expressing *Cas9* in the majority of the clock neurons (B) and flies with both UAS-constructs (D) show bimodal activity with an M anticipation peak around lights-on and an E anticipation peak around lights-off. KO of per using *clk856*-GAL4 (C) abolishes M and E anticipation similar to *per01* flies. (E) Morning and Evening Anticipation are significantly reduced in the *clk856-GAL4* mediated Knockout (MA: $F_{(2,85)}$=18.6895, p<0.0001, p<0.01 for both controls. EA: $F_{(2,85)}$=114.6644, p<0.0001, p<0.01 for both controls). Most importantly, both indices of the knockout strain are indistinguishable from *per01* flies (MA: p=0.6667, EA: p=0.4884) (F–H) Immunocytochemistry of Control (*clk856>Cas9*) and KO (*clk856>Cas9, perG*) staining against PER (magenta) and PDF (cyan). Control flies show PER staining in both, sLNvs and lLNvs, whereas there is no detectable PER signal in the PDF cells in the KO strain (E). Similarly, we see six LNds in the control and two LNds in the experimental flies, showing that some neurons can escape (F) The number of PER+ DN1s is strongly reduced in the KO strain and we do not detect PER in the DN2 neurons (G). (I–L) A clock in LNd or PDF neurons is necessary for rhythmic behavior. *Clk856*-GAL4 mediated KO reduces rhythmicity to less than 10% (H). KO in PDF neurons (*pdf*-GAL4) (I) or in the LNds (*MB122B*-split-GAL4) (J) had no effect on rhythmicity. KO in both places (*Mai179*-GAL4) significantly decreases rhythmicity (K).

DOI: https://doi.org/10.7554/eLife.48301.009

The following figure supplements are available for figure 4:

**Figure supplement 1.** Immunolabeling of cell-specific per KO using *Mai179-GAL4*, *repo-GAL4* and *GMR-ss00681-split*-GAL4.
DOI: https://doi.org/10.7554/eLife.48301.010

**Figure supplement 2.** PER knockout in non-lateral neuron clusters does not affect rhythmicity levels.
DOI: https://doi.org/10.7554/eLife.48301.011

example, there was no detectable nuclear PER signal in all PDF cells or in the DN2s (*Figure 4F and H*). There is also a marked reduction in the number of PER-positive DN1s in the dorsal brain; this is expected as the *clk856*-GAL4 line does not express in all DN1 neurons (*Gummadova et al., 2009*) (*Figure 4H*). Similarly, most LNds are PER-negative. There are however two LNds that remain PER-positive for some reason (*Figure 4G*), that is, there are a few cell escapers. We note that the PERKO strategy is also effective with weaker and more narrowly expressed GAL4 lines (*Figure 4—figure supplement 1*), indicating that it can be used to investigate the contribution of clocks in individual neuron subgroups to circadian behavior.

We next addressed the contribution of clocks within individual neuron subgroups to DD rhythmicity. Previous work assigned a central role to PDF neurons and specifically to the small LNvs: ablating these cells eliminates DD rhythms, and expressing per in these same neurons restores DD rhythms to *per01* flies (*Grima et al., 2004*; *Stoleru et al., 2004*). We were therefore surprised that the PERKO with *pdf*-GAL4 had only a marginal effect on DD rhythmicity compared to the controls (*Figure 4J*). Similarly, a PERKO in the cells important for controlling E activity (E cells: 3 LNds and the 5th sLNv) with *MB122B*-split-GAL4 had no effect on rhythmicity (*Figure 4K*). However, a PERKO in both groups achieved with *Mai179*-GAL4, lowered rhythmicity to less than 20% (*Figure 4L*). Similar results were obtained with *dvPDF*-GAL4, which expresses in similar neuron groups (data not shown).

To address whether other neurons have similar effects, we expressed the PER guides elsewhere: knockout in the retina (*GMR*-GAL4), glial cells (*repo*-GAL4) or DN1s (*clk4.1M*-GAL4 and *AstC*-GAL4) were successful and did not affect rhythmicity (*Figure 4—figure supplement 2*). These findings taken together suggest that a clock in either of two key places, the sLNvs or the LNds, can drive rhythmic behavior.

We also assayed the free-running DD periods of flies lacking PER in individual neuron subgroups (*Figure 5A* and *Table 1*). These periods did not change if the PERKO was in the dorsal brain and/or in the large PDF neurons, the lLNvs. However, a PERKO in the E cells and with drivers expressing in these cells plus some dorsal neurons results in a slight but significant period lengthening of approximately 0.5 hr. In contrast, a PERKO in most of the lateral neuron clusters gave rise to a short period. These two sets of period phenotypes taken together suggest that the clocks in the two different key neuron subgroups collaborate to achieve the intermediate and close to 24 hr period characteristic of wild-type flies.

## Discussion

The central clock of animals is essential for dictating the myriad diurnal changes in physiology and behavior. Knocking out core clock components such as *period* or *Clock* severely disrupts circadian behavior as well as molecular clock properties in flies and mammals (*Allada et al., 1998*;

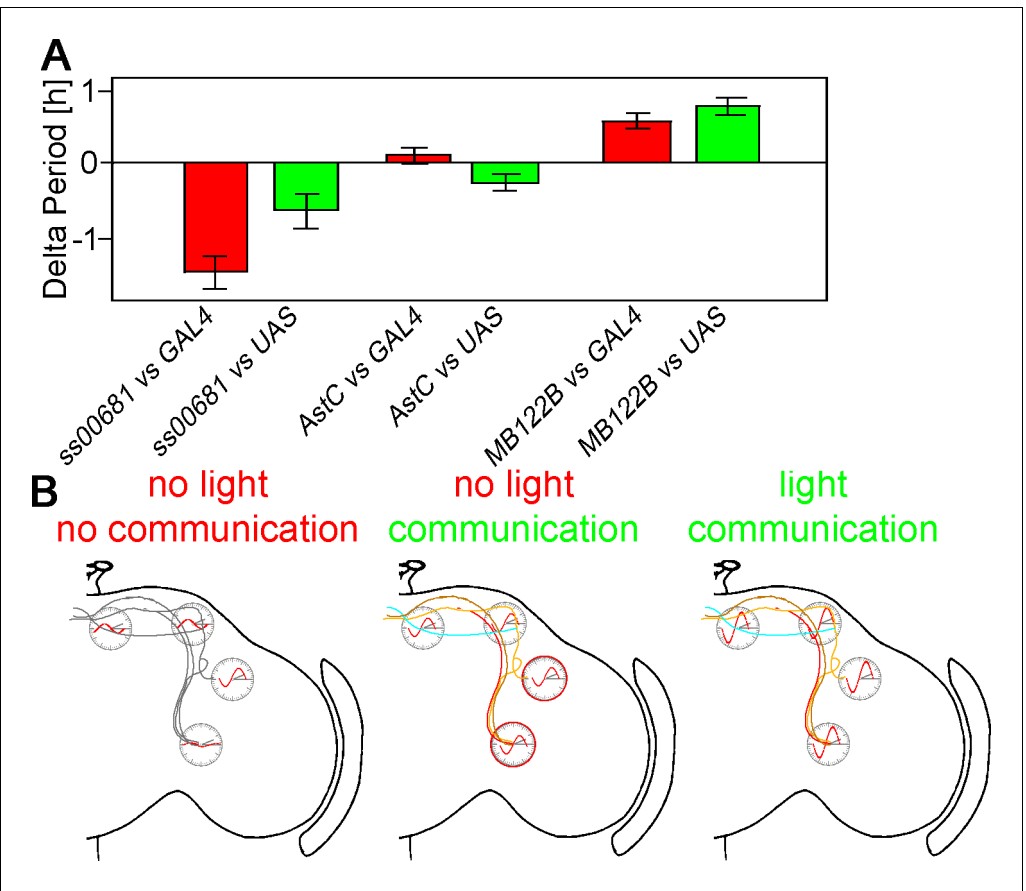

**Figure 5.** KO of PER in subsets of neurons changes free-running period. (**A**) Changes of free-running period upon KO. Red bars represent the change of period between the KO line and the GAL4 control (± SEM), green bars represent the change of period between the KO line and the UAS control (± SEM). (**B**) Model of neuronal communication and light influencing the *Drosophila* clock machinery. Silencing the clock network in DD causes a damping of molecular oscillations and a drifting apart from the common phase as indicated by the red waves. If network communication is allowed, the different neuronal sub-clusters are mostly in sync and show robust cycling, suggesting neuronal communication is essential for molecular oscillations. In a normal LD cycle light drives high amplitude and synchronized cycling even in the absence of neuronal communication, establishing a hierarchy of synchronization cues with light on the top.

DOI: https://doi.org/10.7554/eLife.48301.012

**Table 1.** Changes in free-running period after PER knockout with different GAL4 drivers.

| GAL4 line used | Number of flies | Change compared to GAL4 control | Change compared to UAS control |
| --- | --- | --- | --- |
| *dv-PDF-GAL4* | 6 | −2.41 ± 0.37 | −0.79 ± 0.37 |
| *Mai179-GAL4* | 6 | −1.38 ± 0.68 | −1.07 ± 0.68 |
| *GMR-ss00681* | 12 | −1.39 ± 0.19 | −0.56 ± 0.19 |
| *GMR-ss00367* | 32 | −0.12 ± 0.13 | −0.14 ± 0.13 |
| *AstC-GAL4* | 16 | 0.11 ± 0.11 | −0.26 ± 0.11 |
| *GMR-ss1038* | 24 | 0.06 ± 0.11 | 0.06 ± 0.11 |
| *VGlut-GAL4* | 11 | 0.10 ± 0.11 | −0.26 ± 0.11 |
| *GMR-ss00849* | 22 | 0.55 ± 0.13 | 0.54 ± 0.13 |
| *MB122B* | 21 | 0.57 ± 0.11 | 0.76 ± 0.11 |

DOI: https://doi.org/10.7554/eLife.48301.013

*Gekakis et al., 1998*; *Konopka and Benzer, 1971*). Here we show that similar behavioral effects occur when we silence the central clock neurons and thereby abolish communication within this network and with downstream targets, that is fly behavior becomes arrhythmic in LD as well as DD conditions and resembles the phenotypes of core clock mutant strains (*Konopka and Benzer, 1971*).

Despite the loss of all rhythmic behavior, silencing did not impact the molecular machinery in LD conditions: PER and PDP1 protein cycling was normal. These findings suggest that 1) rhythmic behavior requires clock neuron output, which is uncoupled from the circadian molecular machinery by network silencing, and 2) synchronized molecular rhythms of clock neurons do not require neuronal activity. These findings are in agreement with previous work showing that silencing the PDF neurons had no effect on PER cycling within these neurons (*Depetris-Chauvin et al., 2011*; *Nitabach et al., 2002*; *Wu et al., 2008*). The results presumably reflect the strong effect of the external light-dark cycle on these oscillators.

In DD however, the individual neurons change dramatically: the different neurons desynchronize, and their protein cycling damps to different extents. Interestingly, sLNv cycling relies most strongly on neuronal communication: these neurons cycle robustly in controls but apparently not at all in the silenced state. sLNvs were previously shown to be essential for DD rhythms (*Grima et al., 2004*; *Stoleru et al., 2004*). Unfortunately, the sensitivity of immunohistochemistry precludes determining whether the molecular clock has actually stopped or whether silencing has only (dramatically) reduced cycling amplitude. However, a simple interpretation of the adult-specific silencing experiment favors a stopped clock: decreasing the temperature to 18 degrees after a week at high temperature failed to rescue rhythmic behavior. A similar experiment in mammals gave rise to the opposite result, suggesting an effect of firing on circadian amplitude in that case (*Yamaguchi et al., 2003*). However, we cannot at this point exclude different explanations, for example chronic effects of neuronal silencing or a too large phase difference between the different neuronal subgroups to reverse after a week without communication.

In either case, a stopped clock or an effect on clock protein oscillation amplitude, these results make another link to the mammalian literature: modeling of the clock network suggests that different neurons resynchronize more easily if the most highly-connected cells are intrinsically weak oscillators (*Webb et al., 2012*). The sLNvs are essential for DD rhythms, known to communicate with other clock neurons (*Grima et al., 2004*; *Stoleru et al., 2004*) and are situated in the accessory medulla; this is an area of extensive neuronal interactions in many insects (*Reischig and Stengl, 2003*). These considerations rationalize weak sLNv oscillators.

An important role of interneuron communication in DD is in agreement with previous work showing that altering the speed of individual neuron groups can change the phase of downstream target neurons (*Yao and Shafer, 2014*). An important signaling molecule is the neuropeptide PDF: its absence changes the phase of downstream target neurons, and silencing PDF neurons causes an essentially identical phenotype to the lack of PDF (*Im et al., 2011*; *Lin et al., 2004*; *Wu et al., 2008*). However, the effects reported here are much stronger and show different levels of autonomy than PDF ablation, suggesting that other signaling molecules and/or the neuronal activity of additional clock neurons are essential to maintain proper rhythmic clock protein expression.

To address these possibilities, we took two approaches. First, we investigated clock gene RNA levels after silencing. The goal was to assess whether the damping of silenced neurons is under gene expression control, likely transcriptional control. Indeed, *tim* mRNA profiles nicely reproduced the protein cycling profiles: robust cycling of all (assayed) clock neurons was maintained in LD even with silencing, but *tim*-mRNA levels in the sLNvs stopped cycling in DD; in contrast, robust cycling was maintained in the LNds (*Figure 3C and E*). This suggests that the changes in protein cycling amplitude and also possibly phase are under transcriptional control. Importantly, the *tim* signal in the sLNvs disappeared upon silencing, suggesting that neuronal activity promotes clock gene transcription at least in this subset of neurons. This recapitulates for the first time in *Drosophila* the robust positive relationship between neuronal firing and clock gene transcription in mammals (*Shigeyoshi et al., 1997*). To date, *Drosophila* neuronal firing had only been connected to post-transcriptional clock protein regulation, namely TIM degradation (*Guo et al., 2014*). Conceivably, these two effects are connected: TIM degradation might be required to relieve transcriptional repression and maintain cycling.

The second approach was a cell-specific knockout strategy, applied to the clock neuron network. We generated three guides targeting the CDS of *per* and also expressed CAS9 in a cell-specific

manner. The guides caused double strand breaks in the *per* gene, which in turn led to cell-specific *per* mutations. This adult brain knockout strategy worked reliably and specifically, in glial cells as well as neurons, with high efficiency and with no apparent background effects (*Figure 4B–G*). We have successfully used this strategy to knock out most if not all *Drosophila* GPCRs (data not shown) and believe it will be superior to RNAi for most purposes. Importantly, expression of the guides with the *clk856*-GAL4 driver phenocopied *per*[01] behavior (*Figure 4C and H*). To focus on individual clock neurons, we generated cell-specific knockouts in different clock neurons. PERKO in the PDF cells did not increase the level of arrhythmicity, only a PERKO in most lateral neurons, E cells as well as PDF cells, generated high levels of arrhythmic behavior.

This result is surprising given previous rescue or knockdown experiments (*Herrero et al., 2017*; *Stoleru et al., 2004*): *per* rescue in all neurons except the PDF neurons (*elav-GAL4 pdf-GAL80*) did not restore wildtype behavior. However, this rescue significantly improved the rhythmicity index compared to control flies, which can be interpreted as lower rhythm power, a feature we also observed in our *pdf* PERKO. Knockdown experiments further suggested an importance of the clock in PDF cells. As knockdowns strongly depend on GAL4/RNAi strength, it can generate variable levels of knockdown in different cells. This can cause phase differences among different neurons, which would lead in turn to arrhythmicity. From this point of view, the guide strategy appears to be superior; it mutates DNA sequences and should be more robust to differences in expression strength. This is exemplified by successful PERKO using split-GAL4 lines (*Figure 4—figure supplement 1*).

Even though a clock in the PDF neurons appears to be dispensable for rhythmicity, the cell bodies still appear essential. This is because ablating the PDF cells causes high levels of arrhythmicity (*Stoleru et al., 2004*). The data shown here therefore suggest that the LNds can drive the rhythmic output of key sLNv genes like PDF even in the absence of a clock in these neurons. Consistent with this interpretation, recent data indicate that the LNds project dendritic as well as axonal arborizations into the accessory medulla, the location of the sLNvs, indicating extensive communication between these two important subgroups of clock neurons (*Schlichting et al., 2019b*). This interpretation is further supported by examining the free-running period of the cell-specific knockouts: Ablating *per* in the LNds causes a long period, whereas ablating PER in most lateral clock neurons causes a short period phenotype. These data suggest that interactions between the sLNvs and other clock cells, perhaps within the accessory medulla, are essential for the close to 24 hr speed of the overall brain and behavioral clock.

While this manuscript was being written, we became aware of two other studies addressing the contribution of different circadian subgroups and neuronal interactions to *Drosophila* rhythms. The strategy, results and conclusions in the first study overlap extensively to what we report here (Delventhal et al., cosubmitted paper). That work exploited guides against *tim* as well as against *per* and thoroughly characterized the efficacy of the cell-specific knockout strategy, including effects on mRNA cycling.

The second study is recently published and similarly highlights the dependence of DD rhythms on network properties (*Bulthuis et al., 2019*). However, they report a decrease in rhythmicity upon knockout of the circadian clock in PDF cells, an effect that neither we nor Delventhal observed. This difference may be due to their knockout strategy, namely, overexpression of a dominant negative Cycle isoform within PDF cells. This protein may have effects on gene expression beyond knocking out the circadian clock. The distinction furthermore suggests that the conceptually simpler PERKO strategy has fewer side effects and is therefore superior.

Some of the communication properties described here resemble what has been found in mammalian systems. For example, decreasing neuronal interactions by creating sparse SCN cultures changes the free-running period and activity phase of individual neurons (*Welsh et al., 1995*). This suggests that communication is also critical for circadian phase and period determination in mammals. Nonetheless, fly clock cells may be even less cell-autonomous and more dependent on communication than what has been described for mammals (reviewed in *Evans, 2016*). First, the fly system may be particularly dependent on light. For example, peripheral fly clocks appear strongly light-dependent in contrast to what has been described for mammalian liver (reviewed in *Ito and Tomioka, 2016*). Although much of the fly data could reflect cellular asynchrony in constant darkness, circadian cycling in the periphery crashes rapidly under these conditions and resembles the strong and rapid non-cycling that occurs in the sLNvs upon silencing in DD. Notably, fly cryptochrome but not mammalian cryptochrome is light-sensitive (reviewed in *Michael et al., 2017*) and probably contributes

to the light-dependence of fly peripheral clocks. This is also because light can directly penetrate the thin insect cuticle, which probably contributes to making the fly brain less dependent on ocular photoreception than the mammalian brain. However, some fly clock neurons do not express cryptochrome, suggesting that the fly clock system is dependent on network interactions even in a light-dark cycle (*Benito et al., 2008*; *Yoshii et al., 2008*). These considerations suggest that the fly circadian network is an attractive object of study not only because of its limited size of 75 neuron pairs but also because of its strong dependence on neuronal communication.

# Materials and methods

## Key resources table

| Reagent type | Designation | Source or reference | Identifiers | Additional information |
|---|---|---|---|---|
| Genetic reagent (*D. melanogaster*) | clk856-GAL4 | *Gummadova et al., 2009* | Flybase: FBtp0069616 | |
| Genetic reagent (*D. melanogaster*) | UAS-*Kir2.1* | Bloomington *Drosophila* Stock Center | BDSC_6595 | |
| Genetic reagent (*D. melanogaster*) | pdf-*GS*-GAL4 | *Depetris-Chauvin et al., 2011* | FBal0267534 | |
| Genetic reagent (*D. melanogaster*) | per$^{01}$ | *Konopka and Benzer, 1971* | FBal0013649 | |
| Genetic reagent (*D. melanogaster*) | mai179-GAL4 | *Grima et al., 2004* | FBal0124017 | |
| Genetic reagent (*D. melanogaster*) | MB122B-GAL4 | *Guo et al., 2017* | | |
| Genetic reagent (*D. melanogaster*) | pdf-GAL4 | *Renn et al., 1999* | FBtp0011844 | |
| Genetic reagent (*D. melanogaster*) | clk4.1M-GAL4 | *Zhang et al., 2010* | FBtp0054012 | |
| Genetic reagent (*D. melanogaster*) | UAS-*Cas9.P2* | Bloomington *Drosophila* Stock Center | BDSC_58986 | |
| Genetic reagent (*D. melanogaster*) | AstC-GAL4 | Bloomington *Drosophila* Stock Center | BDSC_52017 | |
| Genetic reagent (*D. melanogaster*) | dvPDF-GAL4 | *Guo et al., 2014* | FBtp0081543 | |
| Genetic reagent (*D. melanogaster*) | w;CyO/Sco; MKRS/TM6B | Bloomington *Drosophila* Stock Center | BDSC_3703 | |
| Genetic reagent (*D. melanogaster*) | VGlut-GAL4 | Bloomington *Drosophila* Stock Center | BDSC_60312 | |
| Genetic reagent (*D. melanogaster*) | GMR-ss00650-GAL4 | G Rubin, Janelia Research Campus | | |
| Genetic reagent (*D. melanogaster*) | GMR-ss01038-GAL4 | G Rubin, Janelia Research Campus | | |
| Genetic reagent (*D. melanogaster*) | GMR-ss00849-GAL4 | G Rubin, Janelia Research Campus | | |
| Genetic reagent (*D. melanogaster*) | GMR-ss00367-GAL4 | G Rubin, Janelia Research Campus | | |
| Genetic reagent (*D. melanogaster*) | GMR-ss00681-GAL4 | *Liang et al. (2019)* | | |
| Genetic reagent (*D. melanogaster*) | GMR-ss00645-GAL4 | G Rubin, Janelia Research Campus | | |

*Continued on next page*

*Continued*

| Reagent type | Designation | Source or reference | Identifiers | Additional information |
|---|---|---|---|---|
| Genetic reagent (*D. melanogaster*) | *GMR*-GAL4 | Bloomington *Drosophila* Stock Center | BDSC_1104 | |
| Genetic reagent (*D. melanogaster*) | *repo*-GAL4 | Bloomington *Drosophila* Stock Center | BDSC_7415 | |
| Genetic reagent (*D. melanogaster*) | *tub*-GAL80ts | Bloomington *Drosophila* Stock Center | BDSC_7018 | |
| Genetic reagent (*D. melanogaster*) | *UAS-per-g* | This study | | |
| Recombinant DNA reagent | pCFD6 | Addgene | RRID: Addgene73915 | |
| Software | Fiji | https://fiji.sc/ | | |
| Software | Stata SE15 | | | |
| Software | Microsoft Office Excel | | | |
| Software | ActogramJ | actogramj. neurofly.de | | |
| Antibody | Anti-PER Rabbit polyclonal | *Stanewsky et al., 1998* | | 1:1000 |
| Antibody | Anti-PDF Mouse monoclonal | Developmental Studies Hybridoma Bank | AB_760350 | 1:500 |
| Antibody | Anti-PDP1 Guinea pig polyclonal | *Benito et al., 2007* | | 1:2000 |
| Antibody | Goat anti-mouse polyclonal | ThermoFisher | A-31575 | 1:200 |
| Antibody | Goat anti-rabbit polyclonal | ThermoFisher | A-11034 | 1:200 |
| Antibody | Goat anti-rabbit polyclonal | ThermoFisher | A-32732 | 1:200 |
| Antibody | Goat anti-guinea pig polyclonal | ThermoFisher | A-11073 | 1:200 |
| Commercial assay or kit | MIDI-Prep Kit | Qiagen | 12143 | |
| Commercial assay or kit | Q5 Polymerase | NEB | M0491S | |
| Commercial assay or kit | BbSI-Enzyme | NEB | R0539S | |
| Commercial assay or kit | Vectashield | Vectorlabs | H-1000 | |

## Fly strains and rearing

All flies were reared on standard cornmeal medium at a temperature of 25℃, with the exception of adult-specific silencing experiments for which flies were raised at 18℃. The SS00849, SS00367, SS01038, SS00645, SS00650 lines were made and characterized by H Dionne and A Nern in the laboratory of G Rubin (Janelia Research Campus).

## Fly line generation

We generated a UAS-*per-g* line following the protocol published by *Port and Bullock (2016)*. In short, we digested the pCFD6 Vector (addgene #73915) with BbsI, PCR amplified two PCR fragments carrying three guides targeting the CDS of *per* and performed a Gibson Assembly to include

those in the pCFD6 backbone. Positive clones were sent for injection to Rainbow Transgenic Flies Inc (Camarillo, CA, USA) and the transgene was inserted into the second chromosome by phi-recombinase using BL 8621. Flies were crossed to $w^{1118}$ for screening and positive individuals were balanced using BL 3703. The following guide sequences were used:

*per* guide1: GGCAGAGCCACAACGACCTC
*per* guide2: CAAGATCATGGAGCACCCGG
*per* guide3: GAGCAAGATCATGGAGCACC

## Behavior recording and data analysis

Individual 2–6 days old male flies were singularly transferred into glass tubes (diameter 0.5 mm) with food (2% agar and 4% sucrose) on one end and a cotton plug to close the tube on the other end. The tubes were placed into *Drosophila* Activity Monitors (DAM, Trikinetics) in a way that the infrared light beam was located in the center of the tube. A computer measured the number of light-beam interruptions caused by the movement of the fly in one-minute intervals.

Standard light-dark to constant darkness experiments: We recorded the behavior of all flies at a constant temperature of 25°C for 5–7 days under standard light-dark conditions of 12 hr light and 12 hr darkness (LD12:12) followed by constant darkness (DD) for at least 6 days. We generated actograms using ActogramJ (*Schmid et al., 2011*). We next generated average activity profiles of the last 3 days of LD condition as previously described (*Schlichting and Helfrich-Förster, 2015*). In short, we averaged the minute-by-minute activity of the last three days for each individual fly. The average activity of individual flies of the same genotype were then averaged and converted into 30 min bins. Anticipation indices were calculated as follows: Morning anticipation index (MA) = sum of activity ZT21-ZT0/sum of activity ZT18-ZT0. Evening anticipation index (EA) = sum of activity ZT9 – ZT12/sum of activity ZT6 – ZT12. Anticipation indices of less than or equal to 0.5 represents no anticipation, whereas a value greater than 0.5 represents anticipatory behavior. Statistical analysis was performed by a one-sample t-test comparing the index to the value of 0.5. Genotypes were compared using one-way-ANOVA with post-hoc Tukey test (three genotypes, Stata SE15) or a student's t-test (two genotypes, Excel).

DD behavior was analyzed using the first six days of constant darkness (DD1-6) using $chi^2$ analysis. Only flies surviving the whole DD period were used. Based on this analysis, we determined dead, rhythmic and arrhythmic flies and calculated the percentage of rhythmic flies. Using the data only from rhythmic flies, we calculated the free-running period and SEM. Statistical analysis was performed using one-sample t-test or a one-way ANOVA with post-hoc Tukey test.

All experiments were performed at least twice using between 25–32 flies per genotype. Only flies surviving the whole LD or DD condition were used for data analysis. Data in this paper represent the first experiment performed.

## Adult-specific silencing experiments

We raised the flies at 18 degrees and performed two separate sets of experiments: In the LD experiment, we recorded the behavior of the flies for 3 days at 18°C and switched to 30°C to silence the neurons and followed the behavioral change within the same sets of flies. We generated average activity at 18°C (days 2–3) and at 30°C (days 4–5) as described above.

In a second set of experiments, we raised the two groups of flies at 18°C and then performed LD to DD experiments either at 18°C or 30°C. Flies were entrained for 3 days and then released into DD. In the 30°C experiment, we decreased the temperature back to 18°C at Circadian Time (CT) 0 after 6 days in DD to investigate possible emergence of rhythmic behavior after silencing. We continued recording the behavior for six more days in DD at 18°C.

## Immunohistochemistry

2–6 days old male flies were entrained in LD 12:12 at 25°C for three days and collected at ZT0 to analyze the CRISPR/Cas9 knockout strategy. To investigate clock protein cycling, 2–6 days old male flies were entrained in LD 12:12 at 25°C for 5 days and collected in 6 hr intervals around the clock. Similarly, flies were entrained for 5 days and released into DD for 5 more days to obtain cycling data at DD5. For adult-specific silencing experiments, flies were raised at 18°C. 2–6 days old male flies were collected and entrained for 5 days in LD 12:12 at 30°C and released in DD at 30°C for 5 days.

The whole flies were fixed for 2 hr 45 min in 4% paraformaldehyde (PFA) in phosphate-buffered saline (pH = 7.4) including 0.5% TritonX (PBST). The flies were rinsed 5 times for 10 min each with PBST and subsequently the brains were dissected in PBST. Brains were blocked in 5% normal goat serum (NGS) in PBST for 3 hr at room temperature (RT). The primary antibody (rabbit anti-PER, 1:1000, *Stanewsky et al., 1998*, mouse anti-PDF, 1:500, *Drosophila* Studies Hybridoma Library (DSHB), C7 and guinea-pig anti PDP1, 1:2000, *Benito et al., 2007*) was applied overnight at RT and the brains subsequently rinsed 5 × 10 mins with PBST. Secondary antibodies (Alexa, Fisher Scientific, 1:200) were applied for 3 hr at RT. Afterwards, the brains were rinsed 5 × 10 mins with PBST and mounted on glass slides using Vectashield (Vector Laboratories INC., Burlingame, CA, USA) mounting medium.

Confocal microscopy was performed using a Leica SP5 microscope. Sections of 1.5 um thickness were obtained. Laser settings were kept constant across genotypes to obtain comparable results. Image acquisition was performed using Fiji. Staining intensity was assessed by quantifying the brightest 3 × 3 pixel area of individual neurons of at least five brains per timepoint. Each experiment consists of at least two biological repeats. Three different background intensities were determined the same way and subtracted from the neuronal intensity. Data points represent average and SEM.

Statistical analysis was performed either by 2-way ANOVA (Stata SE15) to check for time- and genotype dependency. To assess damping behavior, we subtracted the minimum average staining intensity (the lowest point of the cycling analysis) from individual brains from the highest timepoint. These values were compared using one-way ANOVA (statsa) or student's t-test.

### Fluorescent in-situ hybridization (fish)

2–6 days old male flies were entrained for 5 days in LD 12:12 and collected in 6 hr intervals around the clock. In a second set of experiments, flies were released into DD for 5 days and collected in 6 hr intervals. Flies were dissected fresh under red light to avoid phase-shifting the molecular machinery. Brains were subsequently fixed in 4% PFA in PBS for 55 min at RT. Afterwards, brains were washed 3 × 10 min in PBST and dehydrated as described in *Long et al. (2017)*. Brains were kept in 100% EtOH until all time points were collected and all further steps were done simultaneously as described in *Long et al. (2017)*.

A set of 20-probe sequences were designed for the entire *pdf* mRNA sequence and conjugated with Quasar 570 (Stellaris Probes, Biosearch Technologies, CA, USA). The *tim* probes consist of a set of 48-probe sequences against the entire *tim* mRNA sequence, including the 5' and 3' untranslated regions. The *tim* probes were conjugated with Quasar 670 dye (Stellaris Probes, Biosearch Technologies, CA, USA). Probes were diluted to a stock concentration of 25 μM and aliquoted in −20°C. The final concentration of *pdf* probes and *tim* probes were 250 nM and 750 nM, respectively.

Brains were mounted on glass slides using Vectashield mounting medium (Vector Laboratories INC., Burlingame, CA, USA) and scanned using a Leica SP5 microscope in 1.5 um sections. All samples were scanned in one session to avoid signal loss. Fluorescence intensity was assessed by quantifying the brightest 3 × 3 pixel area of individual neurons of at least five brains. Each experiment consists of at least two biological repeats. Three different background intensities were determined the same way and subtracted from the neuronal intensity. Data points represent average and SEM.

## Acknowledgements

We thank Dr. Todd C Holmes, Dr. Lisa M Baik and David Au for discussions and for sharing relevant unpublished data. We also thank Dr. Fang Guo and Dr. Katharine Abruzzi for discussions and comments on the manuscript as well as Dr. Norbert Perrimon and Dr. Paul Hardin for providing fly lines and antibodies. We thank H Dionne, A Nern and G Rubin (Janelia Research Campus) for providing the unpublished split-GAL4 lines: SS00849, SS00367, SS01038, SS00645 and SS00650. Stocks obtained from the Bloomington *Drosophila* Stock Center (NIH P40OD018537) were used in this study. This work was supported by the Howard Hughes Medical Institute (HHMI). MS was sponsored by a DFG research fellowship (SCHL2135 1/1).

## Additional information

### Funding

| Funder | Author |
|--------|--------|
| Howard Hughes Medical Institute | Matthias Schlichting<br>Madelen M Díaz<br>Jason Xin<br>Michael Rosbash |
| Deutsche Forschungsgemeinschaft | Matthias Schlichting |

The funders had no role in study design, data collection and interpretation, or the decision to submit the work for publication.

### Author contributions

Matthias Schlichting, Conceptualization, Data curation, Formal analysis, Investigation, Visualization, Writing—original draft, Writing—review and editing; Madelen M Díaz, Conceptualization, Formal analysis, Investigation; Jason Xin, Investigation; Michael Rosbash, Conceptualization, Resources, Supervision, Funding acquisition, Writing—original draft, Project administration, Writing—review and editing

### Author ORCIDs

Matthias Schlichting [iD] https://orcid.org/0000-0002-0822-0265
Michael Rosbash [iD] https://orcid.org/0000-0003-3366-1780

### Decision letter and Author response

Decision letter https://doi.org/10.7554/eLife.48301.018
Author response https://doi.org/10.7554/eLife.48301.019

## Additional files

### Supplementary files

• Transparent reporting form
DOI: https://doi.org/10.7554/eLife.48301.014

### Data availability

All raw data are deposited on Dryad (DOI: https://doi.org/10.5061/dryad.7s75p25).

The following dataset was generated:

| Author(s) | Year | Dataset title | Dataset URL | Database and Identifier |
|-----------|------|---------------|-------------|-------------------------|
| Schlichting M, Díaz M, Xin J, Rosbash M | 2019 | Data from: Neuron-specific knockouts indicate the importance of network communication to *Drosophila* rhythmicity | https://doi.org/10.5061/dryad.7s75p25 | Dryad Digital Repository, 10.5061/dryad.7s75p25 |

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
