## [Decision Letter]

Thank you for submitting your article "Neuron-specific knockouts indicate the importance of network communication to *Drosophila* rhythmicity" for consideration by *eLife*. Your article has been reviewed by three peer reviewers, and the evaluation has been overseen by a Reviewing Editor and Catherine Dulac as the Senior Editor. The reviewers have opted to remain anonymous.

The reviewers have discussed the reviews with one another and the Reviewing Editor has drafted this decision to help you prepare a revised submission.

Summary:

The manuscript by Schlichting et al. investigates the mechanisms by which clock cells in the *Drosophila* brain generate rhythmic behavior. Using cell-specific CRISPR, the authors show that a clock in ventral lateral neurons (LNds), generally thought of as the central clock cells in *Drosophila*, is not necessary for behavioral rhythms. Loss of rhythms only occurs when clocks are disrupted in ventral and dorsal lateral (LNd) neurons. In addition, molecular circadian cycling in LNvs appears to depend upon network connectivity as it is lost when clocks cells are silenced, even though cycling in LNds is maintained. In general, network connectivity is critical for function in the clock circuit in flies.

Overall, the reviewers found the manuscript of significant interest and noted the potential conceptual advance in our understanding of circadian mechanisms in *Drosophila*. However, the conclusions need to be substantiated by additional experiments:

Essential Revisions:

1) The result that molecular cycling in LNvs (PDF cells) is eliminated when all clock neurons are silenced does not preclude the possibility that silencing in PDF cells alone is sufficient for this effect. If so, the data would not support loss of communication between neuronal groups as the source of the molecular defect in PDF neurons. The authors infer, from previous studies of PDF-cell manipulations, that adult-specific silencing of these cells alone will not disrupt cycling, but a direct comparison is required.

2) The authors claim that abolishing the clock in PDF cells with targeted per CRISPR has no effect compared to controls. However, the *pdf*-GAL4 controls for CRISPR flies show only 40% rhythmicity. This is rather far from the other control, which shows about 95% rhythmicity, and it strongly weakens the conclusion. The results do not allow the conclusion that killing the clock in PDF cells has no discernable effect, although it probably has less effect than killing the clock in LNv + LNd (20% rhythmicity according to Figure 4K). Given that the conclusions here contradict previous findings – for instance, this lab (Stoleru et al., 2004) showed that flies are arrhythmic when PER is rescued in all except PDF cells, and Herrero et al., 2017, showed that targeted per/tim RNAi in PDF cells strongly affects DD rhythms – it is important that the authors repeat their CRISPR experiments in a background with wild type rhythms.

3) The authors state that CRISPR/Cas9 mediated period knockout was 90% efficient, but the basis for this conclusion is not clear. How was this assessed? It is important to document the efficacy of the CRISPR knockout, not only for all the positive data, but also for the negative behavioral results with PER KO in the visual system, glia and DNs.

4) The manuscript lacks information regarding experimental details and statistical analysis. Figure legends do not include relevant information such as the number of animals or brains evaluated, how many times (if any) a particular experiment was repeated, or alternatively, whether what is shown is a representative experiment. Rhythmicity of each line is represented as a percentage plotted in a bar graph, but how was rhythmicity determined and how many flies were tested in each experiment? The average activity plots are not described. Are several days for a single fly plotted or a specific day for a number of flies? Please clarify. In addition, given that control lines show differences in the M/E anticipation peaks, some quantitative measurement of this behavioral clock output is necessary/ highly recommended to compare the relevance of different neuronal clusters to the M/E peak.

---

## [Author Response]

Essential Revisions:1) The result that molecular cycling in LNvs (PDF cells) is eliminated when all clock neurons are silenced does not preclude the possibility that silencing in PDF cells alone is sufficient for this effect. If so, the data would not support loss of communication between neuronal groups as the source of the molecular defect in PDF neurons. The authors infer, from previous studies of PDF-cell manipulations, that adult-specific silencing of these cells alone will not disrupt cycling, but a direct comparison is required.

We repeated once again the experiment described in Figure 2—figure supplement 2 but this time silenced either the whole clock network as before or only the PDF neurons in an adult-specific fashion using *tubGAL80^ts^*. Consistent with our previous results, we found that silencing the whole clock network lead to a strong reduction of cycling amplitude in the sLNvs and DN1s, whereas the LNds were not affected. Silencing the PDF neurons caused a shift of PER staining maximum and only a slight reduction of cycling amplitude in the sLNvs. The LNds were unaffected, and the DN1 neurons changed phase like before. These data confirm the findings we referred to in our first submission and support the notion of network communication being critical for molecular rhythms even in the PDF neurons.

2) The authors claim that abolishing the clock in PDF cells with targeted per CRISPR has no effect compared to controls. However, the pdfgal4 controls for CRISPR flies show only 40% rhythmicity. This is rather far from the other control, which shows about 95% rhythmicity, and it strongly weakens the conclusion. The results do not allow the conclusion that killing the clock in PDF cells has no discernable effect, although it probably has less effect than killing the clock in LNv + LNd (20% rhythmicity according to Figure 4K). Given that the conclusions here contradict previous findings – for instance, this lab (Stoleru et al., 2004) showed that flies are arrhythmic when PER is rescued in all except PDF cells, and Herrero et al., 2017, showed that targeted per/tim RNAi in PDF cells strongly affects DD rhythms – it is important that the authors repeat their CRISPR experiments in a background with wild type rhythms.

We agree that this experiment was not ideal and we exchanged the genetic background of our *pdf*-GAL4 line while the manuscript was under review. We repeated the experiment as suggested. In both repeats the percent of rhythmic flies significantly increased and ranged between 82-90 percent suggesting that the low percentage in our previous results was indeed a genetic background problem. We also want to mention again that Delventhal et al. (this manuscript was submitted back to back with ours) found exactly the same using the guide-based strategy but independent guide sequences.

To address the results from these previous publications (we agree that discussing this is important), we modified our Discussion and included a paragraph discussing the cell-ablation, RNAi and rescue experiments. It is also worth noting that the analysis performed in this paper is based on chi^2^ analysis of 6-7 days of behavior in DD. Many labs use shorter periods of darkness, which makes this way of analysis less powerful and can cause higher levels of arrhythmicity. In addition, we did not distinguish between strong and weakly rhythmic flies as some labs do. If we calculate the power of the rhythm, *pdf*-mediated PER KO has a power of 32 compared to 41 and 44 in the controls, suggesting that the flies’ rhythms are indeed weaker. Delventhal et al. sees a similar trend using a different way of analysis: even though they also see a high level of rhythmic flies, the rhythms are weaker than in controls.

3) The authors state that CRISPR/Cas9 mediated period knockout was 90% efficient, but the basis for this conclusion is not clear. How was this assessed? It is important to document the efficacy of the CRISPR knockout, not only for all the positive data, but also for the negative behavioral results with PER KO in the visual system, glia and DNs.

The value of 90% efficiency is derived from analyzing the lateral clock neurons of the *clk856*-mediated knockout in the lateral neurons, as we know that this driver expresses in all of these cells. To obtain this value of 90%, we counted the lateral neurons of 10 brains. We neglected to state this clearly in this section but do so now.

We also included representative images of a *repo*-KO and a *GMR-ss00681* knockout, which eliminated most of the PER+ glial cells or PER in the sLNvs, respectively. Most of the dorsal neuron drivers (including the split lines) function in poorly defined subsets of DN1 neurons. As we could not label the guide/cas9 expressing neurons, this precluded us from determining efficiency. Similar efficiencies were also found in Delventhal et al. who did a more thorough job in describing the efficacy of the method.

4) The manuscript lacks information regarding experimental details and statistical analysis. Figure legends do not include relevant information such as the number of animals or brains evaluated, how many times (if any) a particular experiment was repeated, or alternatively, whether what is shown is a representative experiment. Rhythmicity of each line is represented as a percentage plotted in a bar graph, but how was rhythmicity determined and how many flies were tested in each experiment?

This should have been done better. Our apologies. We have now improved our figures and included numbers of flies. Each experiment was at least performed twice. We also included a statistics section in Materials and methods and include p-values in the figure legends and Results.

The average activity plots are not described. Are several days for a single fly plotted or a specific day for a number of flies? Please clarify.

The average activity profiles represent the average of the last 3 days in LD condition. The first days were discarded to allow for proper entrainment of the flies. We added this information to Materials and methods.

In addition, given that control lines show differences in the M/E anticipation peaks, some quantitative measurement of this behavioral clock output is necessary/ highly recommended to compare the relevance of different neuronal clusters to the M/E peak.

We analyzed M and E anticipation indices and included them in Figures 1 and 3 to overcome this difficulty.